# Impact of X-Linked Hypophosphatemia on Muscle Symptoms

**DOI:** 10.3390/genes13122415

**Published:** 2022-12-19

**Authors:** Cecilia Romagnoli, Teresa Iantomasi, Maria Luisa Brandi

**Affiliations:** 1Department of Experimental and Clinical Biomedical Sciences “Mario Serio”, University of Florence, Viale Pieraccini 6, 50139 Florence, Italy; 2F.I.R.M.O. Italian Foundation for the Research on Bone Diseases, Via San Gallo 123, 50129 Florence, Italy

**Keywords:** X-linked hypophosphatemia, FGF23, skeletal muscle, muscle weakness, skeletal muscle dysfunction

## Abstract

X-linked hypophosphatemia (XLH) is the most common hereditary form of rickets and deficiency of renal tubular phosphate transport in humans. XLH is caused by the inactivation of mutations within the phosphate-regulating endopeptidase homolog X-linked (*PHEX*) gene and follows an X-dominant transmission. It has an estimated frequency of 1 case per 20,000, and over 300 distinct pathogenic variations have been reported that result in an excess of fibroblast growth factor 23 (FGF23) in the serum. Increased levels of FGF23 lead to renal phosphate loss, decreased serum 1,25-dihydroxyvitamin D, and increased metabolism of 1,25-dihydoxyvitamin D, resulting in hypophosphatemia. Major clinical manifestations include rickets, bone deformities, and growth retardation that develop during childhood, and osteomalacia-related fractures or pseudo-fractures, degenerative osteoarthritis, enthesopathy, dental anomalies, and hearing loss during adulthood, which can affect quality of life. In addition, fatigue is also a common symptom in patients with XLH, who experience decreased motion, muscle weakness, and pain, contributing to altered quality of life. The clinical and biomedical characteristics of XLH are extensively defined in bone tissue since skeletal deformations and mineralization defects are the most evident effects of high FGF23 and low serum phosphate levels. However, despite the muscular symptoms that XLH causes, very few reports are available on the effects of FGF23 and phosphate in muscle tissue. Given the close relationship between bones and skeletal muscles, studying the effects of FGF23 and phosphate on muscle could provide additional opportunities to understand the interactions between these two important compartments of the body. By describing the current literature on XLH and skeletal muscle dysfunctions, the purpose of this review is to highlight future areas of research that could contribute to a better understanding of XLH muscular disability and its management.

## 1. Introduction X-Linked Hypophosphatemia

X-Linked hypophosphatemia (XLH; Online Mendelian Inheritance in Man (OMIM) #307800) is a rare, genetic and progressive musculoskeletal disease caused by mutations that lead to loss of function of the X-linked homologous endopeptidase (*PHEX*) gene (OMIM: #300550) that regulates phosphate [1]. The *PHEX* gene is located in the Xp22.1 chromosome and is strongly expressed in the skeleton. XLH is the most commonly occurring form of inherited low-phosphate rickets with a predicted incidence of 3.9 per 100,000 live births and a prevalence ranging from 1.7 per 100,000 children to 4.8 per 100,000 persons (children and adults) [1,2]. XLH patients are affected by multiple medical complications and present with some degree of disproportionate dwarfism with predominant shortening of the lower extremities, poor mineral density, and rickets or osteomalacia [3].

XLH transmission follows an X-dominant inheritance and has been described with a wide array of mutations, including a large number of cases due to *PHEX de novo* mutations [4]. *PHEX* is predominantly expressed in osteoblasts and osteocytes and encodes for an enzyme that breaks down local SIBLING proteins (small integrin-binding ligand, N-linked glycoproteins), in particular osteopontin. Moreover, the enzyme is responsible for decreasing fibroblast growth factor 23 (FGF23) serum level, therefore affecting its expression [5,6].

Thus far, over 350 distinct pathogen variants have been reported [7,8] that result in FGF23 serum level excess. This subsequently leads to renal phosphate loss, decreased serum 1,25-dihydroxyvitamin D, and increased metabolism of 1,25-dihydoxyvitamin D, and, therefore, to hypophosphatemia [9].

*PHEX* mutations, including deletion, nonsense, missense, frameshift, splice site, and duplication mutations, can affect all 22 *PHEX* exons, and also the sites of intronic splice and the 5′ untranslated region [10,11,12]. The identification of a correlation between patients with mutations in similar genetic loci and their phenotypes would be helpful in elucidating the roles of *PHEX* and FGF23 in XLH pathophysiology. Indeed, specific mutations have been associated with particular XLH manifestations, and efforts have been made to correlate serum levels of FGF23 to different degrees of XLH severity [13]. However, at present, studies searching for genotype–phenotype correlations in XLH patients do not lead to the identification of representative correlations, due to the rarity of the disease and the difficulty to reach statistical significance, as well as the diversity of mutations and the large data sets needed to identify specific correlations between genotype and phenotype [14]. Moreover, XLH members of the same family may have different levels of severity, suggesting that there may be other parameters that alter clinical manifestations of XLH [10].

The main XLH manifestation in infants is rickets, which develops when the patient starts to walk, and osteomalacia, which persists throughout life; they may also have genu valgum or/and genu varum, craniosynostosis, undersized height, dental abnormalities, enthesopathy, and pain [15]. As the spectrum of symptoms can vary, the diagnosis may be made after the age of 2 years and even during adulthood, especially for *de novo* cases [16]. Early identification of XLH is crucial because treating affected children early can help minimize inferior limb malformations and dental abnormalities, together with maximization of growth [17,18].

Adults with XLH accumulate medical problems as a result of childhood disease: small stature, malformations of the lower limbs and craniosynostosis, along with other symptoms, such as osteomalacia, osteoarthritis, pseudofractures, fracture nonunion, and hearing loss. In addition, fatigue is a main symptom in patients with XLH, who also experience increased pain, stiffness, decreased mobility, and muscular dysfunctions [19,20], leading to functional impairment, poor quality of life (QoL), and predisposition to a high risk of falling [21].

Conventional pharmacological treatment of XLH patients consists in administration of oral phosphate supplementation and active vitamin D derivatives (calcitriol or calcitriol analogues) in an attempt to replenish phosphorous levels in serum. Although this treatment generally improves biochemical endpoints and rickets, counteracting the downstream effects of excessive FGF23, it may not correct for the underlying pathogenesis, nor normalize the fasting serum phosphorous. Tolerability issues of frequent dosing with phosphate (three to five times daily) and activated vitamin D (one to three times daily) may reduce adherence to protocol and lead to decreased therapeutic advantages. Moreover, conventional management is related to undesirable adverse events, such as hypercalcemia, hypercalciuria, nephrocalcinosis, cardiovascular abnormalities, and hyperparathyroidism. Consequently, these therapies need frequent follow-up and dose regulation to balance the enhancement of bone mineralization with the risks of adverse events [22].

In 2018, burosumab, a fully human monoclonal antibody that binds to and inhibits excessive FGF23 activity and raises levels of 1,25 dihydroxyvitamin D, was approved by health authorities in the European Union and the USA for the treatment of patients with symptomatic XLH, on the basis of encouraging results of clinical trials [2,23,24,25]. Burosumab supersedes conventional therapy for children with XLH with respect to the severity of rickets, growth, and biochemistries [26].

## 2. Regulation of Serum Phosphorous through FGF23

The *FGF23* gene is situated on chromosome 12 and the intact circulating FGF23 form is a 32 kDa glycoprotein with a conserved terminal amino part that shares homologies with other members of the FGF family and contains a preserved FGF receptor linkage site [27]. The biologically active FGF23 undergoes mucin-type O-glycosylation in the Golgi apparatus of osteocytes by the enzyme N-acetyl galactosaminyl-transferase 3 (GalNT3). This is necessary for the secretion of active FGF23 by blocking its proteolytic cleft by the protease furin. The proteolytic process is responsible for the formation of the inactive FGF23 C-terminal form (15-17 kDa) [28,29].

FGF23 can either act locally or enter the bloodstream to interact with distant cell surface receptors [3]. FGF23 actions in the tissues are mediated by FGF receptors (FGFRs), FGFR1, FGFR2, FGFR3, and FGFR4, and the expression of such receptors differs from one cellular type to another [30].

Each FGFR has an extracellular domain with three immunoglobulin-like domains, a transmembrane domain and an intracellular domain with a split tyrosine kinase domain [31]. The extracellular domains of FGFR1, FGFR2, and FGFR3 are alternatively spliced, where one splice variation within the third immunoglobulin domain generates FGFR isoforms IIIb (expressed predominately in epithelia) and IIIc (expressed in mesenchymal tissues). Alternative splicing in the third immunoglobulin domain is responsible for the FGF specification [32,33]. FGFR1 is the primary receptor through which the FGF23 exerts its renal phosphaturic effect [34].

The protein α-Klotho has been found to be an essential and critical co-factor for FGF23 signaling [35]. It has been shown that Klotho, associated with the membrane, acts with FGFRs (mainly with FGFR1) in proximal tubular cells to form a receptor complex with high affinity to FGF23, enhancing the activation of the receptor mediated by FGF23 [36].

In the healthy population, FGF23 is secreted primarily by osteocytes and partly by osteoblasts, acting as a phosphaturic factor, inhibitor of 1,25 dihydroxyvitamin D, and modulator of parathyroid hormone (PTH) synthesis and secretion [37,38]. 

In XLH, inactivation of the *PHEX* gene by mutations can lead to malfunction of the mechanisms for detecting phosphate in osteocytes, resulting in inadequate production and secretion of FGF23, excessive renal loss of phosphate, and an evident hypophosphatemic effect (Figure 1) [9].

Specifically, FGF23 aims directly at the kidney to increase urine excretion of phosphate by lowering the expression on the cellular surface of sodium-dependent phosphate co-transporters, NaPi-IIa and NaPi-IIc, in the proximal tubule [39]. Moreover, FGF23 decreases the circulating rates of the active form of vitamin D, suppressing the 1-α-hydroxylase (CYP27B1) enzyme in the kidney, which transforms the pre-hormone 25-hydroxyvitamin D into 1,25-dihydroxyvitamin D. In addition, FGF23 enhances 24-hydroxylase (CYP24A1) expression, an enzyme that degrades 1,25 dihydroxyvitamin D into inactive products and, consequently, decreases the intestinal absorption of phosphate mediated by the NaPi-IIb cotransporter [40]. Furthermore, FGF23 interferes with PTH secretion in the parathyroid gland, which adds to the impact of FGF23 on the reduction in the circulating 1,25 dihydroxyvitamin D levels, and further contributes to its phosphaturic actions [41].

## 3. Skeletal Muscle Dysfunctions in XLH

Skeletal deformities and bone mineralization defects are the most evident effects of high FGF23 levels and low serum phosphate. However, XLH also affects muscle function, as is conceivable considering the close relationship between bone and skeletal muscle [42,43]. The importance of this link is fundamental for mechanical loading and physical activity among various other processes. In recent years, the cross-talk between muscle and bone has become even closer, considering that these tissues are endocrine organs able to secrete several factors that can affect each other’s functions [44].

Muscular pain and weakness, tiredness, and physical deconditioning are all symptoms associated with impaired mobility, increased fall and risk of fractures, and reduction of QoL within the XLH population [2,45].

Muscular symptoms reported in XLH patients are therefore routinely assessed in medical practice through specific questionnaires and tools for evaluating physical function. A clinical management approach that incorporates comprehensive evaluation of musculoskeletal health, including muscular analysis, provides better evidence about the disease, monitoring its evolution and treatment functioning.

An online survey performed by Skrinar et al. collected multiple responses to assess pain severity and disability in XLH patients and their interference in everyday life [46]. The study enrolled 232 adults and 90 children (represented by their parents or caregivers), asking them to answer many questions, including diseases manifestation, demography, family, diagnostic and treatment history, and clinical symptoms. The survey included the following patient-performed outcomes: the Western Ontario and McMaster Universities Osteoarthritis Index (WOMAC^®^), the Brief Pain Inventory (BPI), and the 36-Item Short Form Health Survey version 2 (SF-36v2) to assess pain, physical function and Quality of Life (QoL) in adults. Pediatric Outcomes Data Collection Instrument (PODCI) and the 10-Item Short Form Health Survey (SF-10) were used to assess disability and QoL in children. Results evidenced that pain was very common in both adults and children. In particular, muscular pain was reported in 60% of children and 63% of adults, while muscle weakness was declared by 30% of children and 60% of adults, causing the physical limitations characteristic of XLH disability [46].

The observed muscle weakness in XLH can be explained in part by bone pain due to osteomalacia and ineffective transduction of muscular contraction due to leg deformation or impingement syndrome arising from enthesopathy [47].

Since muscle force is directly correlated with bone strength, anomalous skeletal phenotype in patients with XLH can cause skeletal muscle symptoms. In recent years, it has been shown that osteocytes may also target muscle tissue by releasing, in response to shear stress, two factors, prostaglandin E2 and Wnt3a, which support muscle regeneration and function [48,49].

A study by Veilleux et al. showed that muscle weakness is a clinical XLH characteristic [50]. The study enrolled 34 XLH patients, aged between 6 and 60 years, and foresaw the assessment of muscle quality and function at the lower extremities by peripheral quantitative computed tomography (pQCT) and jumping mechanography techniques, respectively. Results highlighted that XLH patients, regardless of age and gender, have lower muscle density and strength in the lower extremities compared to the control group. However, since the study involved relatively young subjects (the oldest was 50-years-old), they speculated that skeletal muscle symptoms could be responsible for premature mobility function decline in XLH patients [50].

The same research group investigated the relationship between bone and muscle in 30 XLH subjects (ages ranging from 6 to 60 years) [51]. Considering the close connection between the above-mentioned tissues, they hypothesized that a lower deficit in XLH muscle function would lead to weaker bones in patients. They tested bone mass and geometry and muscle size by pQCT and muscular function by jumping mechanography technique. Results evidenced that muscular function was lower in the XLH group despite normal muscle size, and patients had higher bone mineral content with respect to the healthy control, contrary to their initial assumption. Moreover, the results showed that age was not a key factor influencing the findings of the study [50,51].

Evidence regarding the low muscle function of XLH individuals can also be attributed to changes in muscle composition. In fact, it has been shown that increased intra- and inter-muscular fat infiltration is associated with low muscle strength and reduced quality of muscle [50,52,53]. In this regard, Goodpaster et al. described computer tomography as a non-invasive technique to evaluate lipid content in skeletal muscle and validated it as a useful method to provide information about composition and function of the muscle [53].

Recently, Orlando et al. performed a study describing physical function and activity in adult XLH patients [45]. They recruited 26 participants (50% male and 50% female) aged 44 ± 16.1 years. Adult patients were clinically evaluated by physical examination, including handgrip strength (which is the most common clinical tool to assess muscular function), six-minute walk test, jump power and short physical performance battery, and they completed the International Physical Activity Questionnaire to assess physical performance. Data obtained confirmed that muscle power is compromised in XLH and this is responsible for physical activity decay. Interestingly, reported findings highlighted that sex and age are not factors that influence physical functioning in XLH patients. However, in the XLH population, musculoskeletal impairment and sarcopenia may appear earlier with respect to healthy people and, therefore, early therapeutic intervention in XLH individuals could preserve and promote good quality of life by maintaining adequate physical function throughout life [45].

Medical management with conventional treatment (oral phosphate supplements and active vitamin D analogues) during early childhood aims to restore phosphate serum levels, in order to reduce bone deformities during skeletal development [44]. Treatment usually starts when XLH is diagnosed and continues at least until the completion of growth. Correction of hypophosphatemia and improvement of bone mineralization bring great benefits to musculoskeletal health with consequent amelioration in muscle function and reduction in muscle weakness [44]. Notwithstanding, improvement of skeletal growth is variable and depends on the age at diagnosis; the effect of conventional treatment may be non-optimal, with some patients unresponsive [17,54].

Furthermore, discontinuation of the therapy in adult patients leads to a return of chronic hypophosphatemia, causing onset of muscle weakness and fatigue and the consequent worsening of physical function [44]. However, lack of robust clinical trial data regarding its efficacy in adulthood and the difficulty to standardize therapy represent major concerns; therefore, guidelines indicate that conventional therapy should only be prescribed to symptomatic adults [2].

Treatment with burosumab offers great hope for XLH patients. Recent studies in XLH children have shown significant amelioration in serum phosphate levels and musculoskeletal health, with reduced osteoarticular and muscular pain and improved physical function [55,56]. Furthermore, phase III clinical trials have evidenced greater clinical improvement in rickets severity in XLH patients treated with burosumab with respect to patients treated with conventional therapy, as well as benefits regarding musculoskeletal symptoms [23,25,26]. However, clinical studies evaluating long-term effects of burosumab treatment in children and adults with XLH are still lacking; therefore, trials are needed to confirm the safety and efficacy of the drug for XLH therapy.

## 4. Effects of Phosphate and FGF23 on Skeletal Muscle

Phosphate plays a vital role in the metabolism of all living cells and is a major substrate for skeletal muscle, where it is stored in the form of organic phosphorous, particularly adenosine triphosphate (ATP) and phosphoryl creatinine. The free Intracellular inorganic phosphorous (Pi) in the muscular cell is around 1–2 mg/dL (3–5 mmol) and is regulated by Pit1 and Pit2 transporters [57,58].

The remarkable effects of phosphorous depletion in humans and animals suggest adverse effects on skeletal muscle, including muscular weakness, rhabdomyolysis, and creatinuria [59,60,61,62]. Prolonged phosphorous deprivation in humans led to anorexia, muscle weakness, and bone pain, symptoms that rapidly improved when hypophosphatemia was corrected [63].

On the other hand, *in vitro* studies on a C2C12 mouse model have shown that the elevation of phosphate levels impairs myoblast differentiation, significantly decreasing myoblast differentiation gene expression [64,65]. Moreover, it has been shown that hyperphosphatemia promotes cellular senescence of culture myoblasts and reduces their proliferative activity [66].

Pi levels in muscle cells are critical for maintaining creatinine phosphate reserves and the functioning of ATP as a source of energy for muscular mechanical activity. Data obtained in a mouse model by Pesta et al. support the hypothesis that decreased muscle ATP synthesis may be caused in part by low blood Pi concentrations, which may explain some aspects of muscle weakness and myopathy observed in patients with hypophosphatemic rickets [67].

In addition, phosphate is vital for muscular contraction. Studies on animals have highlighted that hypophosphatemia affects the composition of skeletal muscle, interferes with the release of calcium, and compromises the transmembrane potential and the structure of mitochondria of skeletal muscle cells [68,69].

Evidence has shown that phosphate integration counteracts skeletal muscle defects in a case of chronic fatigue and muscle weakness in a patient with hypophosphatemic osteomalacia induced by FGF23 [70,71]. It has also been reported that phosphate supplementation reversed muscle weakness caused by vitamin D deficiency in rats, and muscle trembling and weakness in a dog post-surgery [72,73]. These data highlight that a deficiency in the provision of Pi is the main etiological aspect to explain myopathies in osteomalacia [74,75]. However, a study by Ito et al. reports that acute changes in serum phosphate, causing hypophosphatemia, are not associated with muscular weakness [76].

FGF23 works as a phosphaturic factor in XLH patients. Hypophosphatemia, induced by this hormone, is associated with muscle weakness in XLH [70]. It has also been shown that skeletal muscle weakness, pain, and wasting are manifestations observed in patients with tumor-induced osteomalacia (TIO) [77,78,79]. The lack of genetic mutation and skeletal abnormalities in these patients denotes that FGF23 may contribute to the clinical features either directly or via hypophosphatemia. Studies in the literature report that the removal of an FGF23-releasing tumor mass resolved muscular aches in a patient suffering from TIO [77], and the use of an antibody directed against FGF23 enhanced hand grip strength and spontaneous movement in the Hyp mouse [80]. Moreover, plasma levels of FGF23 are correlated with skeletal muscle mass in hemodialysis patients, which may suggest a role for FGF23 in increasing muscular power [81].

The expression of *PHEX* in myocytes may be indicative of a more direct role for FGF23 in XLH muscle symptoms [82]. Moreover, the presence of the four FGFRs in skeletal muscle satellite cells indicates that skeletal muscle could be a target for FGF23, which is known to be a key contributor in the proliferation and differentiation processes across a broad variety of cells and tissues [83].

However, it remains unknown whether elevated FGF23 levels influence the pathophysiology of skeletal muscle. Sato et al. analyzed the *in vitro* effects of the hormone in human mesenchymal stem cells derived from skeletal muscle, showing that FGF23 induces premature senescence in this cellular model via a promotion of the p53/p21 pathway in a Klotho-independent manner [84]. A study by Avin et al. investigated the direct role of FGF23 on skeletal muscle function by assessing markers of cellular proliferation and differentiation in the *in vitro* mouse cellular model C2C12, showing that increased levels of FGF23, under the experimental testing conditions, may not directly alter skeletal muscle myoblast proliferation, myotube development, or contractility, hypothesizing that other endogenous substances may be required to act in concert with FGF23 to promote muscle dysfunction in hereditary hypophosphatemic rickets [85].

Figure 2 illustrates the effects of phosphate and FGF23 findings on the skeletal muscle discussed in this section.

## 5. Conclusions and Future Perspectives

There is significant muscular impairment experienced by XLH patients with low capability of mobility and lack of functional activity. Despite the muscle-related symptoms caused by XLH, the dearth of information regarding the possible implications of FGF23 and phosphate in skeletal muscle tissue and mechanisms of actions leaves open a vast area that further research could fill with new insights regarding the pathophysiology of XLH.

In particular, further basic research studies, including *in vitro* analysis at the molecular and cellular levels, are necessary for a better and more specific comprehension of the dysfunction in XLH skeletal muscle. Additional knowledge regarding the precise mechanism of action of FGF23 and phosphate in human cellular models may lead to the development of new strategies for the prevention and treatment of this disease.

## Figures and Tables

**Figure 1 genes-13-02415-f001:**
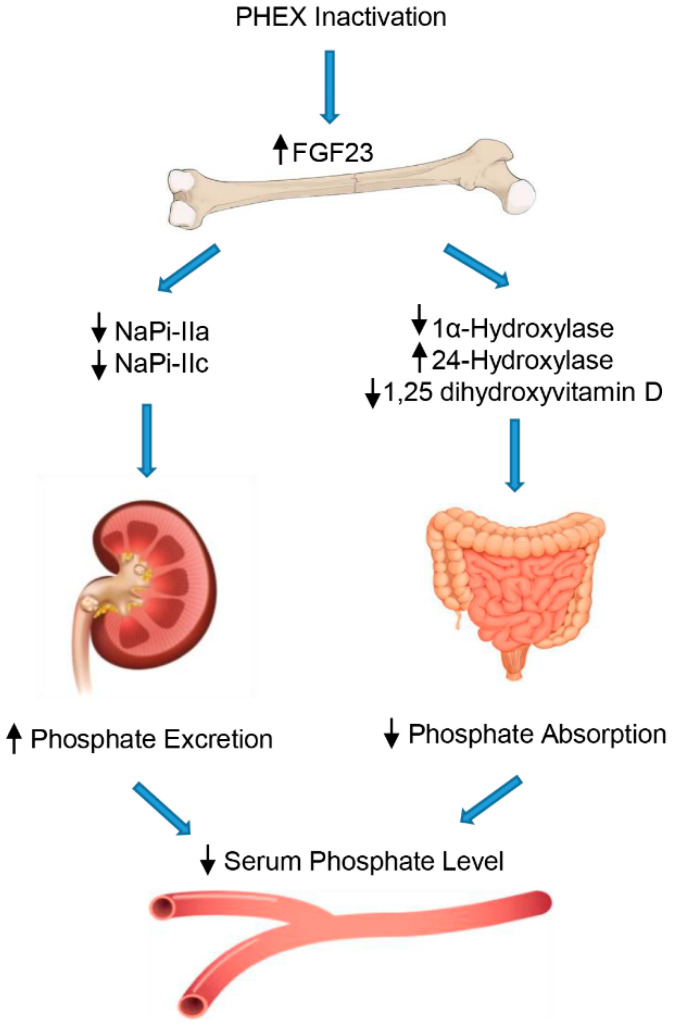
Regulation of phosphate by FGF23. FGF23 decreases the reabsorption of phosphate through regulation of sodium/phosphate cotransporters in the renal proximal tubules. In addition, FGF23 inhibits 1,25-dihydroxyvitamin D production in the proximal renal tubules by downregulating 1α-hydroxylase (CYP27B1) and upregulating CYP24A1 (24-hydroxylase), leading to decreased phosphate absorption in the intestine.

**Figure 2 genes-13-02415-f002:**
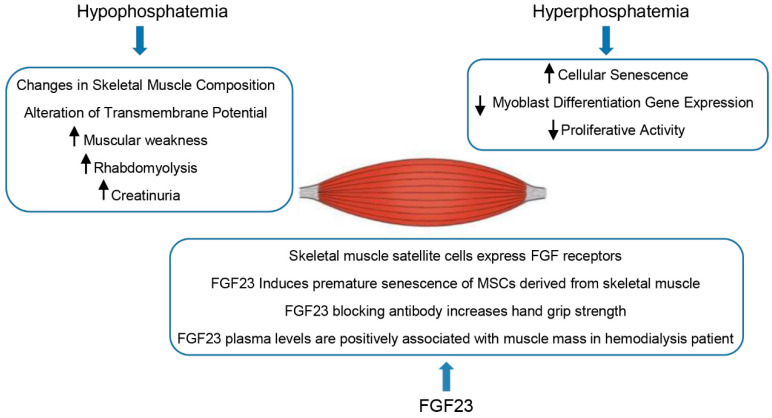
Effects of hypo- and hyper-phosphatemia and FGF23 on skeletal muscle.

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
