# Peer review of "Impact of X-Linked Hypophosphatemia on Muscle Symptoms"

_genes, 2022, doi:10.3390/genes13122415_

Round 1

Reviewer 1 Report

Romagnoli and colleagues provide a detailed review on XLH (X-Linked Hypophosphatemia). The paper is nicely written but I have some concerns.

Since the topic of the review is “skeletal muscle defects”, I would suggest to expand section n. 3 (if there is a limit in word number, section n. 1 could be easily reduced).

How many patients were studied in the main works on muscle health and XLH and by what means (surveys, physical examination, imaging…)? Authors should also consider the paper by Skrinar A et al (doi: 10.1210/js.2018-00365).

Are there significant differences among children and adults?

Is it routine to assess muscle problems in patients and what approach would the authors suggest in clinical management?

Does muscle health improve with current treatments?

Muscoloskeletal health in XLH has been recently reviewed (Glorieux FH et al, https://doi.org/10.1186/s13023-021-02156-x). This work should be cited.

Author Response

Section n.3 was expanded following all the Reviewer’s comments regarding main works, clinical management and treatments and addition of the suggested articles by Skrinar and by Glorieux.

Reviewer 2 Report

“Skeletal Muscle defect in X-linked hypophosphatemia” is a review regarding X-linked hypophosphatemia, the most prevalent cause of hereditary rickets and defect of renal tubular phosphate transport. This review focuses on the impact of XLH on muscular symptoms, a biomedical feature often reported but poorly understood. It then intends to highlight are for future research as an opportunity for better management of XLH and a better understanding of the interaction between muscle and bones. It is an interesting angle with multiple potential outcomes for the clinical and scientific community.

Minor comments:

To facilitate readability, the authors could try to reduce the length of some sentences, by limiting the number of information per sentence to one or two. As an example, in the Abstract, the sentence “It has been estimated...hypophosphatemia” is too long. Or see lines 49-56.

Overall, this manuscript could gain clarity through English proofreading and editing.

Line 49: unnecessary blank space after reference [2].

Line 59: it is unclear what “which includes” refers to

Line 62: Same comment for “their importance”

Lines 64 to 68: That statement is unclear. What type of significant correlations? How a correlation is a cause of rarity?

Lines 71-75: Is it not more common to introduce prevalence and frequency at the beginning? That paragraph could be introduced earlier in that section.

Line 139: Explain what PTH is.

Line 180: The authors mention the “close relationship between bone and skeletal muscle”. More details or a reference would be appropriate.

Lines 184-186: Reference?

Line 235: which results?

Abstract: Italicize gene names (PHEX), line 18

The authors could have put together a Figure to summarize the section “Effects of Phosphate and FGF23 on Skeletal Muscle” as it is the heart of that review.

Reviewer 3 Report

The paper is well-written. The authors offer a complete overview of the FGF23-mediated bone development and remodeling through an analysis of related pathways in physiological and pathological conditions. However, no original/experimental data are reported, and the title appears to be misleading. 

Author Response

The title has been changed to make it more representative of the covered subject.

Round 2

Reviewer 1 Report

The authors addressed my comments